# Identification of an FMNL2 Interactome by Quantitative Mass Spectrometry

**DOI:** 10.3390/ijms25115686

**Published:** 2024-05-23

**Authors:** Sarah Fox, Antoine Gaudreau-LaPierre, Ryan Reshke, Irina Podinic, Derrick J. Gibbings, Laura Trinkle-Mulcahy, John W. Copeland

**Affiliations:** Department of Cellular and Molecular Medicine, Faculty of Medicine, University of Ottawa, Ottawa, ON K1H 8M5, Canada; scopelan@uottawa.ca (S.F.);

**Keywords:** BioID, interactome, formins, FMNL2, extracellular vesicles

## Abstract

Formin Homology Proteins (Formins) are a highly conserved family of cytoskeletal regulatory proteins that participate in a diverse range of cellular processes. FMNL2 is a member of the Diaphanous-Related Formin sub-group, and previous reports suggest FMNL2’s role in filopodia assembly, force generation at lamellipodia, subcellular trafficking, cell–cell junction assembly, and focal adhesion formation. How FMNL2 is recruited to these sites of action is not well understood. To shed light on how FMNL2 activity is partitioned between subcellular locations, we used biotin proximity labeling and proteomic analysis to identify an FMNL2 interactome. The interactome identified known and new FMNL2 interacting proteins with functions related to previously described FMNL2 activities. In addition, our interactome predicts a novel connection between FMNL2 and extracellular vesicle assembly. We show directly that FMNL2 protein is present in exosomes.

## 1. Introduction

Formin Homology proteins (Formins) are a highly conserved family of proteins that regulate cytoskeletal remodeling in a variety of cellular contexts and processes [1,2,3,4,5]. Formins govern cytoskeletal dynamics through two functional domains: Formin Homology 1 (FH1) and Formin Homology 2 (FH2) (Figure 1). FH1 is proline-rich and serves as a ligand for SH3 domains and the small actin-binding protein profilin. FH2 forms a head-to-tail dimer that promotes the polymerization of long, unbranched actin filaments (F-actin) [6,7]. In addition, select FH2 domains also bind and bundle F-actin [8,9,10]. Related to these effects, formins also regulate the organization and stabilization of the microtubule network [11,12,13,14,15]. The fifteen vertebrate formin proteins can be subdivided into families based on the presence or absence of additional regulatory and targeting domains [16,17,18]. These accessory domains govern formin activity, modify their effects on both actin filament polymerization and the organization of higher-order F-actin structures, and impact microtubule dynamics [19,20]. Formins participate in a large range of actin-dependent cellular processes affecting multiple distinct subcellular structures [21,22,23,24,25,26,27]. How formin activity is distributed between these structures is not well understood.

FMNL2 is a member of the Diaphanous-Related Formin subgroup [2] and is predicted to be an essential gene in humans (gnomad.broadinstitute.org). In addition to the actin regulatory FH1 and FH2 domains, the FMNL2 protein also possesses a WH2 domain required for F-actin bundling [8]. Through its effects on F-actin assembly, FMNL2 drives actin-dependent cell motility [28,29,30,31] and participates in diverse cellular functions including force generation at lamellipodia [29,32], filopodia assembly [27,33,34,35], cell–cell junction assembly [36,37], focal adhesion formation [38], and intracellular trafficking [39,40]. FMNL2 activity is regulated by an autoinhibitory interaction between its C-terminal Diaphanous Autoregulatory Domain (DAD) and N-terminal Diaphanous Inhibitory Domain (DID) [8]. Inhibition is relieved by the binding of the appropriate Rho-family GTPase to the GTPase Binding Domain (GBD). FMNL2 has been reported to be an effector of RhoC [31], Cdc42 [29,35,41], and Rac1 [36]. Mutations that affect the DID/DAD interaction render FMNL2 constitutively active [32]. FMNL2 also undergoes N-myristoylation at Glycine residue 2, which is required to target FMNL2 to the plasma membrane [28,29,33,42]. How FMNL2 activity is targeted at other sites of action has not been described. 

Only a handful of studies have been performed to identify the interactomes of specific formin proteins [43,44,45,46]. To gain insight into the mechanisms that regulate FMNL2 activity and subcellular targeting, we used a metabolic labeling (SILAC)-based quantitative BioID approach to generate an FMNL2 interactome. Our screen identified previously described FMNL2-interacting proteins as well as putative FMNL2-interacting proteins that regulate actin dynamics, cell–cell junction assembly, and filopodia and lamellipodia formation, consistent with previously described FMNL2 functions. In addition, we identified several FMNL2 interactors associated with intracellular trafficking and extracellular vesicle release. We show directly, for the first time, that FMNL2 is present in the extracellular vesicles secreted by human melanoma cells. 

## 2. Results

### 2.1. FMNL2 BioID Screen

A challenge in isolating factors associated with cytoskeletal proteins is the harsh conditions required to solubilize the cytoskeletal networks. These conditions are unlikely to preserve transient or fragile protein complexes, thus making affinity purification approaches impractical. Vicinal labeling methods do not rely on the preservation of protein complexes during protein isolation; therefore, harsh conditions can be used to isolate difficult-to-extract cytoskeletal or membrane proteins. To this end, we generated an FMNL2-birA* derivative to facilitate a BioID approach to identifying FMNL2-interacting proteins (Figure 1A). We tested FMNL2-birA* expression in A375 human melanoma cells to determine if the presence of the birA* tag interfered with FMNL2 activity (Figure 1B,C). As described previously, transiently expressed FMNL2-mCherry localizes to the plasma membrane and induces filopodia assembly [27] in A375 cells (Figure 1B). Similarly, FMNL2-birA* also localizes to the plasma membrane and, similar to FMNL2-mCherry, induces filopodia assembly showing that the birA* tag does not obviously interfere with FMNL2 function or subcellular localization. Transfected A375 cells were also incubated overnight with exogenous biotin, causing extensive biotinylation of cellular cytoplasmic proteins as detected by fluorescently labeled streptavidin in cells expressing FMNL2-birA*, but not in cells expressing FMNL2-mCherry (Figure 1B,C). Furthermore, there was extensive colocalization between FMNL2-birA* distribution and streptavidin labeling sites. 

To optimize conditions for BioID, we tested the extent of protein labeling over time in HEK293T/17 cells overexpressing FMNL2-birA*. Cells were transfected with FMNL2-birA* or FMNL2-mCherry as a control. Twenty-four hours post-transfection, the cells were incubated in media supplemented with 50 μM biotin and harvested after an 8- or 16-hour incubation. Total cell lysates were separated on SDS-PAGE, FMNL2-birA* expression was detected by immunoblotting for the encoded myc-tag, and biotinylated proteins were detected separately on a duplicate blot with Streptavidin-HRP (Figure 1D). Under these conditions, the full-length FMNL2-birA* derivative was readily detected at the predicted size and no degradation products were detected. Transitioning from 8 h to 16 h of biotin incubation, there was a clear increase in biotinylation of cellular proteins. No excess biotinylation was detected in control FMNL2-mCherry-expressing cells. We also tested extraction conditions for FMNL2-birA* fusion. Equivalent samples of both whole cell lysates prepared in Laemmli buffer as well as proteins extracted in low salt and high salt RIPA buffers were immunoblotted for FMNL2-birA*. These blots show that the fusion protein was efficiently extracted under both conditions. 

Using these results as a guide for our experimental conditions, we generated an initial FMNL2 interactome using BioID and a Stable Isotope Labeling with Amino acids in Cell culture (SILAC)-based quantitative approach (Figure 1F–H). HEK293T/17 cells transiently expressing FMNL2-birA* were labeled with Heavy SILAC media, while FMNL2-mCherry-expressing cells were labeled with Light SILAC media and served as a control. Twenty-four hours after transfection, cells were incubated in SILAC media supplemented with exogenous biotin for 16 h. Biotinylated proteins were extracted in low-salt RIPA buffer and isolated from cleared cellular lysates by Streptavidin affinity purification. Mass spectrometry identified bound proteins following in-gel trypsin digestion. Putative FMNL2-interacting proteins were ranked based on SILAC ratios twofold above the median and filtered in silico for known background contaminants [47]. Based on these criteria, we identified a high-confidence list of 105 putative FMNL2-interacting proteins (Appendix A).

The function of FMNL2-interacting proteins was assessed by gene ontology analysis. BioProcess analysis revealed significant enrichment for proteins that participate in previously reported FMNL2 activities (Figure 2). As expected, nearly a third of the interactors were involved in regulating actin cytoskeleton assembly [8,29,31]. Other significant categories included the formation of cell–cell junctions [36,37], filopodia and lamellipodia formation [27,28,29,32,33,34], localization of proteins to the plasma membrane [38,39], and vesicle trafficking [38,39] (Figure 2A). Table 1 highlights proteins involved in regulating actin dynamics found in this screening, extracellular vesicle assembly and release, and vesicle fusion. A complete list of the identified interactors is represented in Appendix A. Consistent with the Biological Process compilation, cell compartment analyses of the identified interactors were enriched for the plasma membrane, cell cortex, cell–cell junctions, and cytoplasmic vesicles (Figure 2B). 

### 2.2. Functional FMNL2 Interacting Proteins

As a product of this screen, we identified BAIAP2 and BAIAP2L1 as FMNL2-interacting proteins. The functional relevance of the FMNL2/BAIAP2L1 interaction for filopodia assembly was also recently reported [27]. Similarly, as shown previously, the small GTPase RhoC binds FMNL2 [31] and RhoC was also found on our high-confidence list of interactors. Additional proteins reported to bind FMNL2, including α-catenin [37] and cortactin [48], were also found in this screen. As further proof of principle, we confirmed the cortactin and α-catenin interactions by testing FMNL2-birA*’s ability to biotinylate coexpressed GFP-tagged versions of these proteins (Figure 3A). Both FMNL2-birA* coexpressed with GFP (Figure 3B) and expression of the protein of interest alone served as negative controls (Figure 3A). For both cortactin and α-catenin, co-expression of FMNL2-birA* clearly induced biotinylation of both proteins, but not GFP alone. Neither protein was biotinylated in the absence of FMNL2-birA* (Figure 3A). Using the same strategy, we also validated additional interactors representing distinct aspects of known FMNL2 function and subcellular localization. These interactors included δ-catenin, a component of the adherens junction complex [49,50,51]; the membrane-binding protein RAI14, a sensor of membrane bending [52,53]; PHACTR4 a membrane- and G-actin-binding Protein Phosphatase 1-targeting subunit [54]; and the kinases CDC42BPB and MARK2, both regulators of cell motility [55,56,57,58]. 

### 2.3. FMNL2 and Extracellular Vesicles

We were surprised to note that several putative FMNL2 interactors were associated with extracellular vesicle (ECV) assembly, trafficking, and release, including the synaptobrevin proteins VAMP2, VAMP3, and YKT6 [59,60,61,62,63,64,65,66,67]. Previous reports suggested that each of these proteins is required for ECV production as well as cargo loading into ECVs, suggesting that FMNL2 might also be targeted at these structures. FMNL2’s interaction with each of the three proteins was validated by coexpression with FMNL2-birA* as before (Figure 4A,B), confirming their initial identification in our BioID screen. We then sought to determine if the endogenous FMNL2 protein is present in ECVs released by melanoma cells. A375 melanoma cells were cultured in ECV-free media and after 16 h, the media was collected, cellular debris was removed by low-speed centrifugation, and the released exosomes were purified by ultra-centrifugation [68] (Figure 5). The purified exosomes were quantitated and characterized by nanoparticle tracking analysis (NTA) and shown to have an average diameter of 100 nm stereotypical of these particles. Equivalent whole-cell lysate and exosome samples were subjected to SDS-PAGE and immunoblotted for the cytoplasmic mitochondrial marker TOM20 and the exosome markers TSG101 and Flotillin2. As expected, TOM20 was only detected in the whole-cell lysate, while the exosome markers TSG101 and Flotillin2 were strongly enriched in the purified exosome sample. FMNL2 was also clearly enriched in the exosome sample, indicating that it is contained in these extracellular vesicles. 

## 3. Discussion

In this article, we describe the first FMNL2 interactome generated by BioID. The BioID system has many virtues that make it ideal for characterizing protein interactors associated with the cytoskeleton or cellular membranes. Chiefly, the method does not rely on maintaining the integrity of multi-protein complexes that may not survive the extraction conditions required to solubilize membrane- or cytoskeleton-associated proteins. Indeed, a similar strategy to that described here was used recently to identify Myosin X interactors [69].

The FMNL2 interactome assembled by BioID included the previously described FMNL2-binding proteins α-catenin, cortactin, and RhoA/C. FMNL2 and α-catenin interact during Rac1-induced adherens junction assembly [37]. Cortactin binds FMNL2 to enhance invadopodia formation [48], and the small GTPase RhoC binds FMNL2 to regulate amoeboid invasive cell motility [31]. As a result of this screen, we showed that the I-BAR protein BAIAP2L1 (IRTKS) is an FMNL2-binding protein required for FMNL2-induced filopodia assembly in melanoma cells [27]. In addition to known functional FMNL2 interactors, our screen identified additional novel putative FMNL2-interacting proteins that act in processes associated with known FMNL2 activities and cellular functions as described in the literature and captured by GO analysis (see Table 1 and Appendix A). FMNL2 is thought to induce membrane bending [27,35] and factors associated with membrane bending in its interactome, including SEPTIN2, SEPTIN7, SEPTIN9 [70], BAIAP2, BAIAP2L1 [71], and RAI14 [52]. Similarly, regulators of actin dynamics, including CTTN [48], ENAH [72], MARCKSL1 [73], PHACTR4 [54], and ARPIN [74], were also identified as FMNL2 interactors. Downstream of its effects on actin dynamics, FMNL2 is implicated in regulating cell–cell junction assembly [36,37]; consistent with this, the junctional proteins MLLT4, CTNNB1, CTNND1, JUP, PVRL2 (Nectin-2), and PVRL3 (Nectin-3) were present in our interactome. FMNL2 is also part of the melanoma cell adhesome [75] and promotes β1-integrin trafficking [38]. Accordingly, focal adhesion proteins FERMT2 [75,76] and β1-integrin were also identified as FMNL2-interacting partners. In summary, these findings support the functional relevance of FMNL2 protein–protein interactions identified in our BioID screen.

To our surprise, we identified several FMNL2-interacting proteins involved in regulating vesicle trafficking and membrane fusion. The sorting nexins SNX1, SNX2, SNX5, and SNX9 contain lipid-binding PHOX (aka PX) domains that share FMNL2’s affinity for phosphoinositides. SNX1/2 and SNX5/6 form heterodimers and participate in endosomal recycling, cargo sorting, and retrograde transport [77], while SNX9 is also involved in clathrin-mediated endocytosis [77]. Also related to membrane fusion and intracellular trafficking, we identified SNAP23, SNAP29, and the synaptobrevins VAMP2, VAMP3, and YKT6 as FMNL2 interactors. SNAP23 and SNAP29 are part of the vesicle fusion machinery acting in conjunction with synaptobrevins and syntaxins to mediate vesicle fusion at target membranes [78]. These interactions are consistent with a recently reported role for FMNL2 in intracellular trafficking [39] and suggest FMNL2’s association with the vesicle fusion machinery.

In addition to the roles of VAMP2, VAMP3, and YKT6 in vesicle fusion, these proteins are also involved in extracellular vesicle (ECV) cargo loading and ECV release [64,67,79,80]. FMNL2 activity has recently been associated with exosome release downstream of an EGFL6 signaling pathway [81], and previous proteomic screens found FMNL2 in exosomes released by melanoma cells [82]. Our results confirm these reports and show, for the first time, that FMNL2 is enriched in melanoma cell exosomes. These findings raise intriguing questions regarding FMNL2’s function in ECVs. Given FMNL2’s previously described roles in intracellular trafficking, FMNL2 may also participate in ECV cargo loading, assembly, and release. In addition, our data clearly show that FMNL2 is itself an ECV cargo protein. It will be interesting to determine whether FMNL2 released in ECVs is taken up by target cells and how it might affect target cell behavior, morphology, and motility. 

## 4. Materials and Methods

### 4.1. Reagents and Plasmids

Subcloning of full-length human FMNL2 cDNA into pEF-mCherry was previously described [8,28]. FMNL2 was subcloned into pBirA*-N1 using standard techniques. EGFP-RAI14 and EGFP-PHACTR4 were generated in the Trinkle–Mulcahy laboratory. mEmerald-LASP1-C10 (LASP1) was a gift from Michael Davidson (Addgene plasmid #54141; http://n2t.net/addgene:54141 (accessed on 3 April 2024); RRID:Addgene_54141). pDONR223-MARK2 was a gift from William Hahn and David Root (Addgene plasmid #23404; http://n2t.net/addgene:23404 (accessed on 3 April 2024); RRID:Addgene_23404). pEGFP-N1-Cdc42BPB was a gift from Naoki Mochizuki (Addgene plasmid #50759; http://n2t.net/addgene:50759 (accessed on 3 April 2024); RRID:Addgene_50759). mEmerald-Alpha1-Catenin-C-18 was a gift from Michael Davidson (Addgene plasmid #53982; http://n2t.net/addgene:53982 (accessed on 3 April 2024); RRID:Addgene_53982). pGFP Cortactin was a gift from Kenneth Yamada (Addgene plasmid #50728; http://n2t.net/addgene:50728 (accessed on 3 April 2024); RRID:Addgene_50728). pEGFP VAMP2 was a gift from Thierry Galli (Addgene plasmid #42308; http://n2t.net/addgene:42308 (accessed on 3 April 2024); RRID:Addgene_42308). pEGFP VAMP3 was a gift from Thierry Galli (Addgene plasmid #42310; http://n2t.net/addgene:42310 (accessed on 3 April 2024); RRID:Addgene_42310). YKT6 cDNA HsCD00418102 was obtained from the Harvard Plasmid Repository and subcloned in pEGFP-N1 using standard methods. The following antibodies were used in this study: rabbit anti-GFP (SC8334, SCBT), mouse anti-FLAG (F7425, Sigma, Burlington, MA, USA), rat 9E1 anti-myc antibody (ChromoTEK, Planegg, Germany), rat 5f8 anti-RFP antibody (ChromoTek, Planegg, Germany), mouse 4A10 anti-TSG101 (GeneTex, Irvine, CA, USA), rabbit CA2A3 anti-flotillin2 (Cell Signaling, Danvers, MA, USA), donkey anti-mouse 488, donkey anti-rabbit 488, donkey anti-rat 488, donkey anti-mouse 594, donkey anti-rabbit 594, donkey anti-rat 594 (Jackson Immuoresearch, West Grove, PA, USA), and anti-FLAG-HRP (A8592, Sigma, Burlington, MA, USA). Streptavidin agarose (N1000-005, Vector Laboratories, Newark, CA, USA), GFP-trap beads (Chromatek, Planegg, Germany), and anti-DYKDDDK affinity resin (L00432, Genscript, Piscataway, NJ, USA) were also obtained.

### 4.2. Cell Culture, Transfections, and Treatments

A2058 (CRL-1619) melanoma cells and HEK293T/17 (CRL-11268) cells were obtained from ATCC and cultured in DMEM (Wisent) supplemented with 10% FBS (ATCC) in 5% CO_2_ according to the supplied guidelines. Mycoplasma contamination was tested bi-weekly. Transient transfections were performed using PEI, as described previously [27]. Briefly, 2 × 10^6^ HEK293T/17 cells were seeded into 35 mm dishes 24 h prior to transfection, 1.5 μg total plasmid DNA was diluted in 50 μL Optimem, 7.5 μL of 1 mg/mL PEI was added, and the mixture was incubated for 25–30 min at room temperature. The DNA/PEI mix was added to cells in 1 mL of Optimem and left for 5 h under normal culture conditions. After 5 h, the media was replaced with 2 mL of the appropriate culture medium.

### 4.3. Immunofluorescence

A375 human melanoma cells were prepared for immunofluorescence, as described previously [27]. Briefly, cells cultured on acid-washed glass coverslips were fixed for 10 min directly in 4% paraformaldehyde freshly prepared in PHEM buffer. Following fixation, the cells were permeabilized and blocked for 20 min in 0.3% Triton-X-100 and 5% Donkey Serum (DS) in 1 × PBS. The coverslips were washed in 1 × PBS and incubated with rat anti-myc (9E1, Chromotek, Planegg, Germany) in 0.03% Triton-X-100 and 5% DS in 1 × PBS for 1 h at room temperature. The coverslips were washed 3 times in 1 × PBS and then incubated with a secondary antibody in the same solution for 1 h at room temperature. After washing in 1 × PBS, the coverslips were rinsed in ddH_2_O, mounted in Vectashield, and sealed with nail polish. All microscopy was performed on a Zeiss AXIO Imager.Z1 with a Zeiss Apotome.2 structured illumination system for optical sectioning using a 63× (NA 1.4) Plan Apochromat oil immersion lens and a Zeiss AxioCam HRm camera (60N-C 1” 1.0X 426114) controlled with Zeiss AxioVision (release 4.8.2). Coverslips were mounted in Vectashield (Vector Laboratories, Newark, CA, USA) with or without DAPI. Figures were prepared in Adobe Photoshop v25.7 and Adobe Illustrator v28.5.

### 4.4. BioID Screen

HEK293T/17 cell density and the total amount of FMNL2-birA* expression plasmid were titrated to establish optimal transfection conditions yielding the highest transfection efficiency, as determined by immunofluorescence, overall FMNL2-birA* expression, and immunoblotting for the encoded myc-tag. Immunoblotting of total cellular lysates prepared directly in 1× Laemmli buffer was also used to determine the full length of expressed FMNL2-birA* fusion protein and that no significant degradation products were produced in the transfected cells. 

To establish optimal biotin labeling with the FMNL2.birA* fusion proteins, cells were incubated with 50 μM biotin for 8 and 16 h, 24 h post-transfection. Total cell lysates were prepared in 1× Laemmli buffer, subjected to SDS-PAGE, and blots were probed with Streptavidin-HRP in 10 mg/mL BSA and 0.2% Tween-20 prepared in 1 × PBS. A similar approach was used to test protein extraction conditions. Following biotin labeling, cellular proteins were extracted in high-salt (500 mM NaCl) and low-salt (150 mM NaCl) RIPA buffer (50 mM Tris pH7.4, 1% triton, 0.5% Na-Deoxycholate, 0.1% SDS, 5 mM EDTA, antipain and protease inhibitor cocktail).

We used quantitative stable isotope labeling for amino acids with a culture (SILAC)-based metabolic labeling approach to map the FMNL2 interactome using BioID. HEK293T/17 cells were differentially labeled with media containing either environmental forms of the essential amino acids arginine and lysine (Arg0Lys0, Light media), or isotopic forms (Arg6Lys4, Medium; Arg10Lys8, Heavy) (Cambridge isotope Laboratories, USA). FMNL2-birA* (“bait”) or FMNL2-mCherry (control) fusion proteins were expressed by transient transfection in “Heavy” and “Light” labeled cells, respectively. Twenty-four hours after transfection, 50 μM Biotin (Sigma, St. Louis, MA, USA) was added to the media and the cells were harvested 16 h later in ice-cold, low-salt RIPA buffer. They were then incubated on ice for 15 min to facilitate cellular lysis, followed by 4 rounds of sonication for 10 s. The cellular lysates were cleared by centrifugation for 15 min at 20,000× *g*. Total protein concentrations were measured using the Pierce BCA Protein Assay Kit (Thermo-Fisher, Waltham, MA, USA). Equivalent amounts of total protein extract for Heavy and Light lysates were incubated with Streptavidin-agarose beads (Thermo-Fisher, USA) at 4 °C for 4 h. The beads were washed once in RIPA buffer; then, the beads from the control and experimental pulldowns were combined followed by 3 additional washes. Bound proteins were eluted with 2% SDS/30 mM biotin. Eluted proteins were reduced and alkylated by treatment with DTT and iodoacetamide, respectively. Sample buffer was then added, and the proteins were resolved by electrophoresis on a NuPAGE 10% BisTris gel (Thermo-Fisher, USA). The gel was stained using SimplyBlue Safestain (Thermo-Fisher, USA). The entire lane was cut into 10 slices, with each slice cut into 2 × 2 mm fragments, destained, and digested overnight at 30 °C with Trypsin Gold (Promega, Madison, WI, USA).

Samples of each tryptic digest were analyzed by LC-MS/MS on an Orbitrap Fusion Lumos system (Thermo-Fisher, Waltham, MA, USA) coupled to a Dionex UltiMate 3000 RSLC nano HPLC. The raw files were searched against the Human UniProt Database using MaxQuant software v1.5.5.1 (http://www.maxquant.org; accessed on 10 November 2022) [83] and the following criteria: peptide tolerance = 10 ppm, trypsin as the enzyme (two missed cleavages allowed), and carboxyamidomethylation of cysteine as a fixed modification. Variable modifications are methionine oxidation and N-terminal acetylation. Heavy SILAC labels were Arg10 (R10) and Lys8 (K8). Quantification of SILAC ratios was based on razor and unique peptides, with a minimum ratio count of 2. The peptide and protein FDR was 0.01. These ratios reflect the relative amount of protein captured in experimental (FMNL2-birA*) vs. control (FMNL2-mCherry) experiments. The full dataset is provided in Appendix A.

### 4.5. Gene Ontology

Biological Process: Using Cytoscape (v3.10.0) with the ClueGO plugin (v2.5.10), CluePedia (v1.5.10), and the Biological Process ontology, access numbers of mass spectrometry hits (105 hits) were analyzed for enrichment of Gene Ontology (GO) terms [84,85]. To increase stringency, we used the following parameters: All experimental evidence codes, GO term fusion, *p*-value ≤ 0.05, GO level between 3 and 8, and a minimum of 3 hits per GO term, with said hits being at least 5% of the total gene associated with the GO term. We used the Benjamini–Hochberg correction method for the automatic statistical analysis performed in ClueGO. Twenty-four hits were not annotated with any GO terms.

Cell Compartment: Using Cytoscape (v3.10.0) with the ClueGO plugin (v2.5.10), CluePedia (v1.5.10), and the Cellular component ontology, access numbers of mass spectrometry hits (105 hits) were analyzed for enrichment of Gene Ontology (GO) terms. To increase stringency, we used the following parameters: All experimental evidence codes, GO term fusion, *p*-value ≤ 0.05, GO level between 3 and 8, and a minimum of 3 hits per GO term, with said hits being at least 4% of the total gene associated with the GO term. We used the Benjamini–Hochberg correction method for the automatic statistical analysis performed in ClueGO. Fourteen hits were not annotated with any GO term.

### 4.6. BioID Validation

Putative FMNL2-interacting proteins were filtered in silico for known background contaminants and prioritized based on gene function. Candidate interactors were tested for FMNL2-birA*’s ability to biotinylate the coexpressed protein when transiently expressed in HEK293T/17. Following overnight incubation with exogenous biotin, the coexpressed proteins were purified by immunoprecipitation (IP), as previously described [27]. Briefly, transfected cells were scraped from their dish, washed three times in 1 × PBS, and lysed on ice for 20 min in Co-IP buffer (50 mM Tris pH 7.0, 150 mM NaCl, 1 mM EDTA, 5 mM NaF, 0.5% Triton-X-100, protease inhibitors). Lysates were cleared by centrifugation (10 min, 16,000× *g*) and the supernatant was incubated with anti-DYKDDDDK agarose beads (GenScript, L00432) or GFP-trap beads (Chromotek, gta) for 2 h at 4 °C. The beads were washed three times in Co-IP buffer and the bound proteins were eluted in 1× Laemmli loading buffer. Immunoblotting detected epitope tags for bound proteins. Protein biotinylation was detected with Streptavidin-HRP.

### 4.7. Extracellular Vesicle Isolation

Extracellular vesicles (ECV) were purified by differential ultracentrifugation according to established protocols [68]. A375 human melanoma cells were cultured in ECV-free media (pre-cleared by ultracentrifugation) overnight. The media supernatant was decanted into clean centrifuge tubes and the adherent cells were harvested by trypsinization, pelleted, and lysed in 1 × IP buffer (50 mM Tris pH 7.0, 150 mM NaCl, and 1 mM EDTA, 0.5% Triton-X-100, 5 mM NaF) to generate a soluble cellular lysate. The decanted media was centrifuged at 300× *g* for 10 min to pellet cells and then the supernatant was collected. The cleared supernatant was centrifuged at 2000× *g* for 10 min to pellet dead cells; then, the supernatant was collected and centrifuged at 10,000× *g* for 30 min to remove cellular debris. The resulting supernatant was centrifuged at 100,000× *g* for 70 min to pellet exosomes. This pellet was washed in PBS and the exosomes were re-pelleted at 100,000 g for 70 min. The exosome preparation was characterized by nanoparticle tracking analysis on a ZetaView Nanoparticle Tracking Instrument (ParticleMetrix, Ammersee, Germany). Exosome samples were diluted in PBS to create an acceptable measurement range (typically 1:1000–1:100,000). A total of 1 mL of the sample was injected into the machine and allowed to slow down according to the built-in particle drift sensor. Sample video acquisition and analysis were performed using the following parameters. Acquisition Parameters: sensitivity (85), shutter speed (40), frame rate per second (30), resolution (Highest), and positions measured (11). Post-acquisition parameters: minimum brightness (15), minimum size in pixels (10), and maximum size in pixels (500). For immunoblotting, exosome lysates were prepared in 1 × IP buffer. The protein concentration in the exosome and soluble cell lysates was determined using the Micro BCA kit (Thermo). Equivalent amounts of protein were subjected to SDS-PAGE and blotted for the indicated proteins.

## 5. Conclusions

Using BioID and quantitative mass spectrometry, we describe an FMNL2 interactome for the first time in this article. As a result of this screen, our laboratory previously demonstrated the functional relevance of the BAIAP2L1/FMNL2 interaction for filopodia assembly. Other groups have shown functional interactions between FMNL2 and other putative interactors including α-catenin and cortactin, which were identified in this study. These data validate our screen and its potential for identifying FMNL2 interactors critical to its cellular function. Our results show for the first time that FMNL2 is enriched in exosomes released by melanoma cells and highlight the potential for FMNL2 as a regulator of extracellular vesicle release.

## Figures and Tables

**Figure 1 ijms-25-05686-f001:**
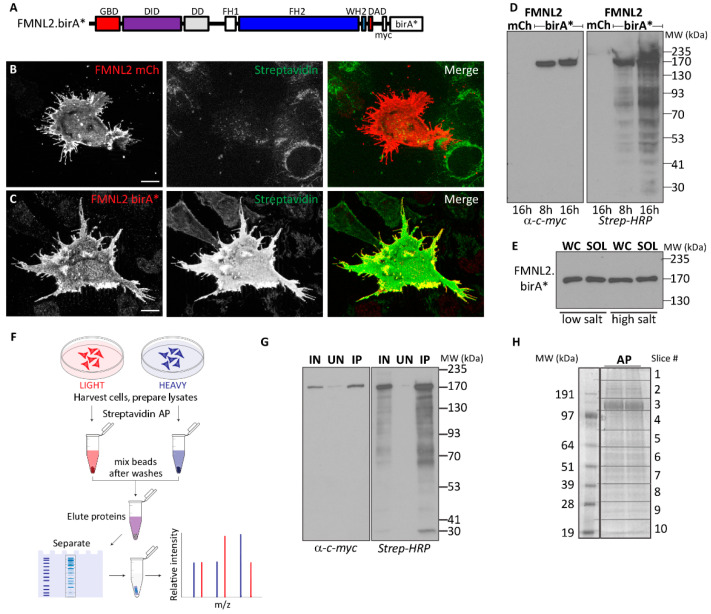
(**A**) Schematic of FMNL2-birA* fusion protein. FMNL2 has multiple regulatory and functional domains: GTPase Binding Domain (GBD), DAD Interacting Domain (DID), Dimerization Domain (DD), Formin Homology 1 (FH1), Formin Homology 2 (FH2), Wasp Homology 2 (WH2), and Diaphanous Autoregulatory Domain (DAD). The position of the myc epitope tag between FMNL2 and birA* is also indicated. (**B**) Transient expression of FMNL2-mCherry (red) induces extensive filopodia formation, but not biotinylation of cellular proteins, in A375 melanoma cells. (**C**) Expression of FMNL2-birA* (red) induces extensive filopodia formation in A375 melanoma cells as well as biotinylation of cellular proteins as revealed by Streptavidin (green) staining in A375 cells. (**D**) FMNL2-birA* degradation products are not detected by immunoblotting for the encoded myc tag in total cell lysates harvested following 8 h or 16 h of cell labeling in media supplemented with exogenous biotin. FMNL2-birA*, but not FMNL2-mCherry, induces extensive biotinylation of cellular proteins after 16 h of cell labeling in the presence of exogenous biotin. (**E**) Full-length FMNL2-birA* is efficiently solubilized in cellular extracts prepared in low-salt and high-salt RIPA buffer. Equivalent samples of whole cell lysates (WC) and solubilized cellular proteins (SOL) were immunoblotted for myc-tagged FMNL2-birA*. (**F**) Workflow for interactome identification using SILAC quantitative mass spectrometry. (**G**) Biotinylated proteins from FMNL2-birA*-expressing cells were isolated by streptavidin affinity chromatography. Soluble Lysate (IN), unbound proteins (UN), and eluted proteins (IP) were immunoblotted for myc-tagged FMNL2-birA* or probed with Streptavidin-HRP. (**H**) Biotinylated proteins isolated from FMNL2-birA* expressing cells were run on preparative SDS-PAGE and visualized with Simply Blue protein stain. Gel slices were prepared as indicated and the extracted proteins were identified by tryptic peptide analysis.

**Figure 2 ijms-25-05686-f002:**
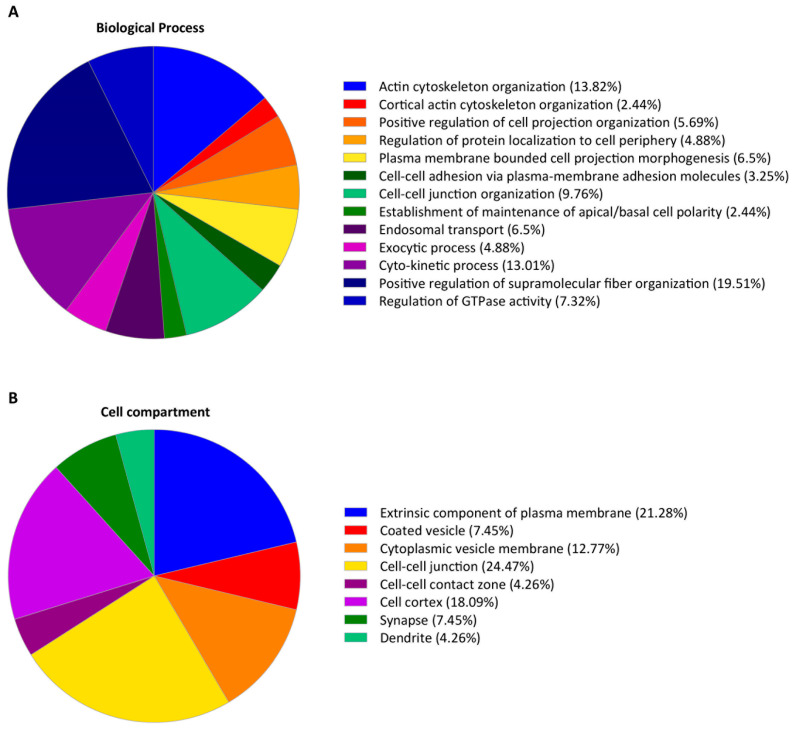
Gene Ontology (GO) analysis of the top candidate proteins identified in the FMNL2 BioID assay. (**A**) Regulation of cytoskeletal dynamics and organization of the plasma membrane were among the main GO terms for Biological Processes associated with the 105 top FMNL2 interacting proteins. (**B**) The plasma membrane, cell cortex, cell–cell junctions, and vesicle membranes were the main GO terms for Cell Compartments of the putative FMNL2 interactors. GO analysis was performed with Cytoscape (v3.10.0) with the ClueGO plugin (v2.5.10) and CluePedia (v1.5.10).

**Figure 3 ijms-25-05686-f003:**
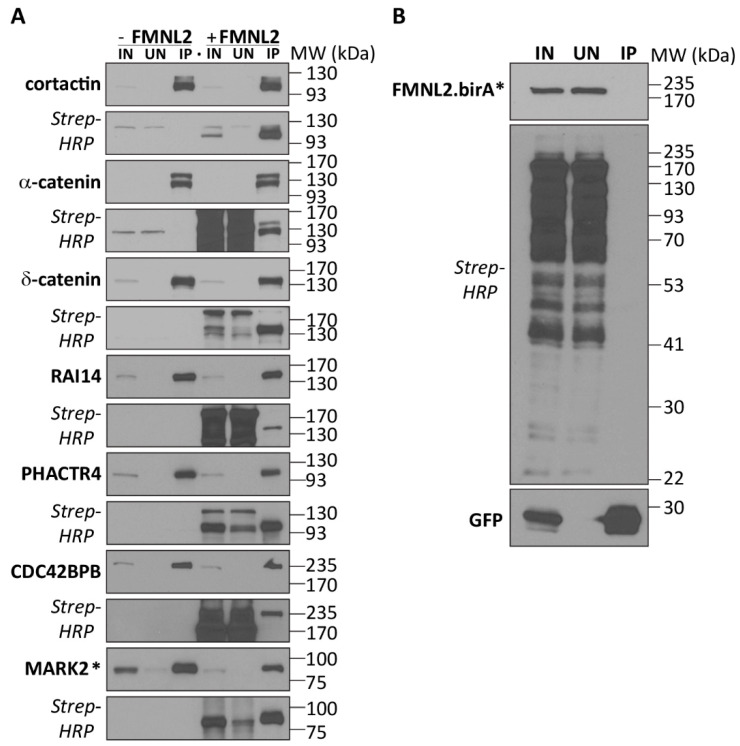
Validation of putative FMNL2-interacting proteins. (**A**) GFP-tagged derivatives of putative FMNL2-interacting proteins were expressed alone (-FMNL2) or coexpressed with FMNL2-birA* (+FMNL2) in HEK293T/17 cells. The GFP-tagged proteins were immunoprecipitated with GFP-trap beads. Equivalent samples of cleared lysates (IN), unbound proteins (UN), and immunoprecipitated proteins (IP) were immunoblotted for the GFP tag. MARK2 was immunoprecipitated via the encoded FLAG-tag and detected by immunoblotting for the FLAG epitope tag. Duplicate blots were probed with Streptavidin-HRP to detect FMNL2-birA*-induced biotinylation of the immunoprecipitated proteins. (**B**) FMNL2-birA did not biotinylate the coexpressed GFP protein. Samples were prepared, immunoprecipitated, and immunoblotted as in (**A**).

**Figure 4 ijms-25-05686-f004:**
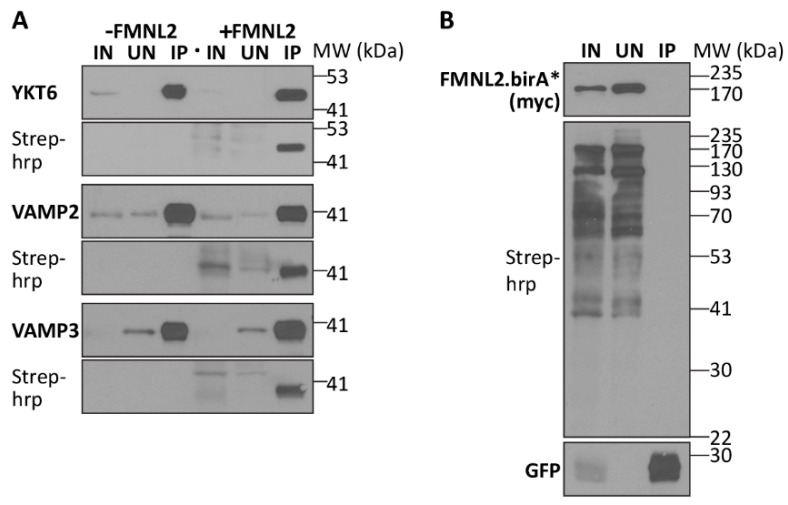
Validation of putative FMNL2-interacting proteins associated with extracellular vesicle release. (**A**) GFP-tagged derivatives of the putative FMNL2-interacting proteins YKT6, VAMP2, and VAMP3 were expressed alone (−FMNL2) or coexpressed with FMNL2-birA* (+FMNL2) in HEK293T/17 cells. The GFP-tagged proteins were immunoprecipitated with GFP-trap beads and equivalent samples of the cleared lysates (IN), unbound proteins (UN) and immunoprecipitated (IP) proteins were immunoblotted for the GFP tag. Duplicate blots were probed with Streptavidin-HRP to detect FMNL2-birA*-induced biotinylation of the immunoprecipitated proteins. (**B**) FMNL2-birA does not biotinylate co-expressed GFP protein. Samples were prepared, immunoprecipitated, and immunoblotted as in (**A**).

**Figure 5 ijms-25-05686-f005:**
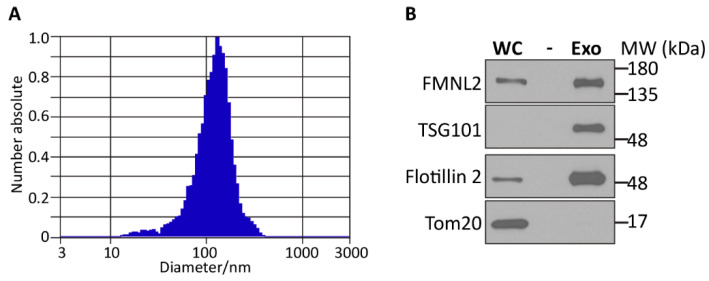
FMNL2 is present in exosomes secreted by melanoma cells. (**A**) Purified exosomes from A375 cells were characterized by nanoparticle tracking analysis. An ECV diameter of 100 nm is typical of exosomes. (**B**) Equivalent amounts of whole-cell lysate (WC) and exosome lysate samples were immunoblotted for the indicated proteins. The mitochondrial marker TOM20 is only detected in whole-cell lysates while the exosome markers Flotillin 2 and TSG101 are enriched or exclusive to the exosome sample. FMNL2 is also enriched in the exosome sample.

**Table 1 ijms-25-05686-t001:** FMNL2-interacting proteins associated with specific cellular functions are identified by SILAC BioID. For each candidate, the number of peptides and unique peptides identified is noted, along with the SILAC ratio quantified by MaxQuant and the Total Intensity. “Bait only” indicates that the protein was only biotinylated/captured from cells expressing birA*-tagged FMNL2. Previously described FMNL2-interacting proteins are indicated with an asterisk. Proteins in bold text were further validated by BioID/Western blot analysis (see Figure 3 and Figure 4 and. [27]). Proteins that appear in the Gingras lab’s BioID control datasets [47] are indicated as follows: (-) not detected, (<) fewer or a similar number of peptides than detected in control datasets.

Protein Class	UniProt	Gene	Peptide	Unique Peptide	SILAC Ratio	Total Intensity	Bkgd
**ECVs**	Q15836	**VAMP3**	5	2	22.22	1.16 × 10^8^	-
	Q9UQN3	CHMP2B	4	4	Bait only	2.28 × 10^7^	-
	P63027	**VAMP2**	5	2	Bait only	9.39 × 10^6^	-
	O15498	**YKT6**	2	2	Bait only	9.16 × 10^6^	-
	Q9NZZ3	CHMP5	2	2	3.09	8.72 × 10^6^	-
	O75955	FLOT1	2	2	3.12	2.65 × 10^6^	-

**Actin**	Q14247	**CTTN ***	13	13	Bait only	1.39 × 10^8^	<
**Cytoskeleton**	Q14008	CKAP5	20	20	3.16	1.16 × 10^8^	<
	Q8N8S7	ENAH	4	4	47.86	3.21 × 10^7^	-
	P13797	PLS3	3	3	2.08	2.59 × 10^7^	-
	Q9UQB8	BAIAP2 *	4	4	7.62	2.27 × 10^7^	-
	Q9UJU6	DBNL	4	4	22.28	1.53 × 10^7^	-
	P49006	MARCKSL1	3	3	Bait only	1.18 × 10^7^	-
	Q7Z6K5	ARPIN	2	2	Bait only	7.17 × 10^6^	-
	Q9UHR4	BAIAP2L1 *	2	2	Bait only	6.05 × 10^6^	-
	Q05682	CALD1	2	2	Bait only	2.86 × 10^6^	-
	Q3V6T2	CCDC88A	2	2	Bait only	2.56 × 10^6^	-
	P61586;P08134	RHOA/C *	2	2	Bait only	1.10 × 10^6^	-

**SNAREs**	Q13596	SNX1	13	11	Bait only	2.94 × 10^8^	-
**and SNXs**	O60749	SNX2	13	11	Bait only	1.19 × 10^8^	<
	Q9BSJ8	ESYT1	3	3	3.51	2.80 × 10^7^	-
	O00161	SNAP23	2	2	Bait only	1.86 × 10^7^	-
	Q9Y5X1	SNX9	6	6	69.76	1.64 × 10^7^	-
	Q9Y5X3	SNX5	5	5	3.54	1.60 × 10^7^	-
	O95721	SNAP29	3	3	Bait only	9.11 × 10^6^	-

## Data Availability

MS data are available from the corresponding authors upon reasonable request.

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
