# Peer review of "Identification of an FMNL2 Interactome by Quantitative Mass Spectrometry"

_ijms, 2024, doi:10.3390/ijms25115686_

Round 1

Reviewer 1 Report

Comments and Suggestions for Authors

Comments and Suggestions:

The article “Identification of an FMNL2 interactome by Quantitative Mass 2 Spectrometry” by Fox et. al., provides an insight about the FMNL2 interactome using BioID and quantitative mass spectrometry. Also, they showed that FMNL2 is enriched in exosomes released by melanoma cells and functions as key regulator in the release of extracellular vesicles.

Clarity and Structure: The introduction effectively sets the stage by presenting thorough explanation about Formin Homology proteins (formins) and FMNL2 and highlighting its significance in different cellular processes and subcellular structures.

Language, Grammar and References: Review the text for any grammatical errors and ensure that the language used is clear and concise. Please add some more relevant references other than self-citations in methods sections.

I do have some minor comments:

1.  Results: The authors are suggested to divide the result section into different sub-sections to have a better view and readability of the outcomes.

2. Page 9, Line 302: The µg and µL symbol should be made correct throughout the manuscript.

3.  Section 4.3: please write about which protein you are targeting for immunofluorescence. Which cells were used for this?

4.      Section 4.5: please add reference for all software’s and plugins used.

5.      Section 4.7: please add reference for EVs isolation.

Author Response

Minor Comments

  • Subheaders have been added to the results section
  • All formatting errors have been corrected in the Methods section
  • Cell type and protein detected have been indicated in the section on immunofluorescence
  • References for the analysis software have been added
  • References for the EV isolation protocol have been added.

Reviewer 2 Report

Comments and Suggestions for Authors

Fox et al. study the interactome of the formin homology family protein FMNL2 by BioID proximity labeling and mass spectrometry, using a SILAC setup for relative quantitation of proteins between those enriched with the birA construct and an mCherry negative control. The differentially enriched proteins are characterized according to Gene Ontology, and selected candidates are verified. Among the putative interactors, several proteins known to be present in extracellular vesicles were identified. Follow-up experiments confirm that FMNL2 is also present in exosomes released from the cell line model under study.

Overall, the experiments are clearly described, and particular attention is paid to appropriate controls and validation experiments. The exosome connection is a nice additional finding that will be of interest to the community. The main drawback of the manuscript is that the mass spectrometry data are only poorly documented, which needs to be improved. Apart from that, I have only minor comments.

Major comments:

Line 136: It is insufficient to show only a network diagram of all putative interactors as the only representation of mass spectrometry data. As a minimum, the protein-level output of MaxQuant should be provided, together with the quantitative information (SILAC ratios) for all proteins. Ideally, all data would be submitted to a proteomics data repository.

Line 360-: More experimental details about the LC-MS setup should be provided, e.g., details about the column used, gradients, flow rates, MS resolution etc.

Minor comments:

Figures of gels: Please add mass units (kDa) to the marker masses for all figures showing gels/blots.

Table 1: The authors should consider cleaning up the table a bit. SILAC ratios are shown with different numbers of significant digits. Total intensities can be shown in exponential format to avoid the many zeros.

Section 4.1: It does not make sense to mention a gift from a co-author. All the information on plasmids could be better presented in a table.

Section 4.2: Some special formatting is wrong, presumably after pasting text into the template: CO2 - CO2, 2x106 - 2x106, symbol for microgram (also in section 4.4).

Line 337: "protein inhibitor cocktail" should probably read "protease inhibitor cocktail" - please specify which one.

Line 351 and elsewhere: 4C should read 4 °C.

Line 359: I think Trypsin Gold is a brand name of Promega. Please clarify.

Line 362: HPLC.

Line 367: Quantitation

Sections 4.6 and 4.7: Centrifugation speeds are stated inconsistently. 16,000 x g, 2000g etc.

Line 414: Please provide experimental details about the nanoparticle tracking (e.g., instrumentation).

References: Some references lack article/page number details: #4,25,33,34,53,69,72

Author Response

We thank the reviewer for their careful reading of the manuscript and have revised it as follows in accordance with their comments.

Major Comments

  • We have included an Excel file as supplementary file 1 that includes the requested information from MaxQuant.
  • We have updated the Materials and Methods with the requested experimental conditions.

Minor comments.

  • All gel figures have been updated with MW (kDa).
  • Table 1 has been modified as requested.
  • The plasmids from the Trinkle-Mulcahy lab are no longer listed as gifts.
  • Special character formatting has been corrected.
  • “Protein” has been corrected to “protease”.
  • Degree symbol has been added where needed.
  • Trypsin Gold supplier has been corrected to Promega
  • HPLC has been corrected
  • Quantitation has been changed to quantification
  • Centrifuge speeds are listed in a consistent format e.g 2000g
  • Additional detail has been provided for the nanoparticle tracking including the instrument and conditions used.
  • References have been amended with page number or doi information.

Round 2

Reviewer 2 Report

Comments and Suggestions for Authors

The authors have addressed all my comments appropriately. I have no further requests.